# Applying Neural Networks in Aerial Vehicle Guidance to Simplify Navigation Systems

**Raúl de Celis \*, Pablo Solano and Luis Cadarso** 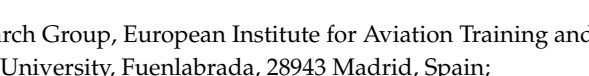

Aerospace Systems and Transport Research Group, European Institute for Aviation Training and Accreditation (EIATA), Rey Juan Carlos University, Fuenlabrada, 28943 Madrid, Spain; pablo.solano@urjc.es (P.S.); luis.cadarso@urjc.es (L.C.)

**\*** Correspondence: raul.decelis@urjc.es; Tel.: +34-914888775

 

**Abstract:** The Guidance, Navigation and Control (GNC) of air and space vehicles has been one of the spearheads of research in the aerospace field in recent times. Using Global Navigation Satellite Systems (GNSS) and inertial navigation systems, accuracy may be detached from range. However, these sensor-based GNC systems may cause significant errors in determining attitude and position. These effects can be ameliorated using additional sensors, independent of cumulative errors. The quadrant photodetector semiactive laser is a good candidate for such a purpose. However, GNC systems' development and construction costs are high. Reducing costs, while maintaining safety and accuracy standards, is key for development in aerospace engineering. Advanced algorithms for getting such standards while eliminating sensors are cornerstone. The development and application of machine learning techniques to GNC poses an innovative path for reducing complexity and costs. Here, a new nonlinear hybridization algorithm, which is based on neural networks, to estimate the gravity vector is presented. Using a neural network means that once it is trained, the physical-mathematical foundations of flight are not relevant; it is the network that returns dynamics to be fed to the GNC algorithm. The gravity vector, which can be accurately predicted, is used to determine vehicle attitude without calling for gyroscopes. Nonlinear simulations based on real flight dynamics are used to train the neural networks. Then, the approach is tested and simulated together with a GNC system. Monte Carlo analysis is conducted to determine performance when uncertainty arises. Simulation results prove that the performance of the presented approach is robust and precise in a six-degree-of-freedom simulation environment.

**Keywords:** nonlinear-flight-mechanics; neural networks; guidance, navigation, and control; machine learning; model; matlab-simulink

## 1. Introduction

Global Navigation Satellite Systems (GNSS) signals are widely utilized for aerospace applications. However, reliability decreases as the requirement of the application for which it is designed increases. The main cause producing this effect is the reduced signal/noise relationship caused by the attenuation and loss of the GNSS signal. This basically means that independent sources of data for navigation are needed to ameliorate these negative effects and reduce interference. Inertial Navigation Systems (INS) are a good example of devices which are independent of external perturbations. Particularly, an inertial estimation unit (IMU) is an electronic gadget that measures and reports a body's specific force, angular rate, and orientation, utilizing a blend of accelerometers, gyroscopes, and sometimes magnetometers. IMUs are normally used in airplanes, including unmanned aeronautical vehicles (UAVs), and spacecraft. However, these systems also feature important lacks, such as frequent incorrect initialization, accelerometer and gyroscope imperfections, which are trigger for cumulative errors

and imperfections in implemented dynamics model. Despite of this fact, Inertial Navigation Systems, when hybridized with GNSS receivers to minimize drift, are excellent for GNC data acquisition [1,2].

However, precision and cost are counterposed objectives. Reducing costs, while maintaining safety and accuracy standards, is key for development in aerospace engineering. Advanced algorithms for getting such standards while cutting down costs are are cornerstone. For example, to maintain an acceptable precision level while reducing costs, less precise devices may substitute expensive systems as long as GNSS signal is reachable and persistent to update the inertial system. However, many scenarios feature high uncertainty and alternatives are needed. An option to satisfy accuracy needs and budget limitations is to merge data of a few low cost sensors, which makes possible increases in accuracy levels.

The advantages of coordinated combination of information have appeared in numerous air applications [3]. For example, information combination strategies for six degrees of freedom rockets are depicted in [4]. The main issues in using various sorts of INS augmented with GNSS updates have been considered by [5]. Notwithstanding INS/GNSS hybridization, a set of nonlinear observers are presented by [6]. Note that, in case there are various sensors available, they may be additional contributions to a filter, e.g., the Kalman filter [1,2].

As it is shown in [1,2], the need to develop new Guidance, Navigation and Control (GNC) frameworks has fostered research on stability and controllability of aerospace vehicles. A novel guidance law is presented in [7], where only observations of line-of-sight angle and its rate of change coming from a seeker are employed. Ref. [8,9] present GNC cooperative techniques based on the conventional Proportional Navigation (PN). In [10] a target-follower engagement is considered, in which the target is followed while it tries to prevent interception. An attitude control-framework device for a spinning sounding rocket, which depends on a proportional, integral, and derivative (PID) controller, is created in [11]. Proportional-derivative GNC laws for the terminal phases of flight are proposed in [12,13]. In [14], a limited time concurrent sliding-mode GNC law is introduced. An overall scheme concerning the guidance and autopilot modules for a class of spin-stabilized balance controlled devices is introduced in [15].

Yet, even in GNSS/IMU hybrid devices, there exist negative influences, such as irregular estimations, which might be predominant during terminal guidance. Other methods, which are based on image recognition using multispectral cameras and other sensors, may be used in navigation for aerospace applications [16]. However, they usually feature high costs. Hence, advancement on new algorithms which may easily fulfill the required precision levels and budget limitations is a foundation in research. For instance, there are recent advances which consist of incorporating IMU, GPS, and laser guidance capacity, offering high accuracy and all-weather capacity [17,18].

Laser guidance may be provided by means of Semi Active Laser Kits (SAL). These devices are applied in many designing areas, such as calculating rotational speed of objects and estimating dynamics of laser spots [19,20]. The bonus of these kits is their favorable position during the last periods of the guidance, when they can provide high precision for GNC systems.

Therefore, it can be stated that sensor hybridization techniques [16,21] for viable and robust estimations are a current need when autonomy, accuracy, and minimal cost are to be achieved. However, also note that as the number of sensors to employed increases, the cost of system also increases. In this sense, Machine Learning (ML) techniques come onto the scene. They offer multitudinous options and innovative solutions of particular interest for GNC applications, where their foray is still latter and shallow, yet with no doubt promising. The utilization of ML strategies for the estimation of parameters dependent on the dynamics of aerial vehicles presents the bit of leeway that once the algorithm is calibrated or trained, it is not important to know the physical-mathematical establishments that rule the flight mechanics. Given the input signals, ML algorithms may restore the data that can later be utilized within the GNC system, such that the subsequent solutions will fit the genuine output [22,23]. Taking benefit of these facts, a reduced set of sensors may be selected to work together

with ML algorithms, all while safety and accuracy standards are matched, and complexity and costs are decreased.

However, the application of these strategies to a wide set of scenarios, which may also include uncertain conditions, depends largely on the representativity and amount of input and output data employed for training ML algorithms. This fact implies that desired performance stability and convergence is to be restricted to the trained mission envelope. Other approaches, which could ensure convergence and stability parameters under the proposed uncertain conditions, might also be employed for this type of application. For instance, adaptive control that uses adaptation laws to online estimate unknown system parameter variations for various mission envelopes [24–26].

Altogether, the objective of this paper is to improve current guidance strategies applying a powerful hybridization approach, which also introduces ML to enable attitude determination with a reduced availability of sensors, namely GNSS, accelerometer and semiactive laser quadrant photo-detectors. In particular, neural networks (NN) are implemented to precisely estimate the gravity vector to be combined with velocity and line of sight vectors in order to determine the attitude or rotation of the vehicle without needing gyroscopes. Note that the mentioned vectors need to be obtained in two different reference frames because otherwise the attitude determination problem cannot be solved.

*Contributions*

The main contribution of this scientific research is the application of Machine Learning techniques, i.e., neural network (NN) algorithms, to hybridize GNSS, accelerometer and semiactive laser quadrant photo-detectors signals. The role of the neural networks is to predict the gravity vector to estimate the attitude of the vehicle. Consequently, the advantage of such a hybrid system over the traditional ones, which are usually based on GNSS and IMUs, is the capability of eliminating gyros, which may be expensive and too sensitive for high demanding maneuvers and not reliable at all during some stages of flight.

The presented approach relies on neural networks and training algorithms to predict the gravity vector in body fixed axes while the vehicle is flying. The three components of the acceleration of the vehicle in body fixed axes are the inputs for the NN. After that, the predicted gravity vector is processed together, by means of a hybridisation algorithm, with velocity and line of sight vector to determine body rotation or attitude.

To reproduce the flight dynamics of an aerial vehicle, a nonlinear mathematical model is proposed, which considers nonlinear aerodynamic forces and moments and that has been validated to build up realistic conditions for simulation experiments [1,2]. On top of that, a robust double-input double-output control algorithm is employed to manage coupling among the normal and lateral nonlinear dynamics.

Note that the presented approach depends on the amount of available data for training, which means stability and convergence may be restricted to the trained mission envelope. However, note that training has been performed for a wide variety of launching, flight, and destination point conditions to resemble realistic settings, i.e., for a comprehensive set of missions. Overall, the methodology results in good enough quality results, even including good response to uncertainty in several conditions and characteristics, i.e., showing good GNC performance. Therefore, the presented research poses a path for a generalized and systematic application of NN/Machine Learning in GNC systems.

The rest of this paper is organized as follows. In Section 2, the system modeling is described in detail. Section 3 describes the navigation, guidance and control algorithms. Section 4 exposes simulations results. Finally, discussion and conclusions are presented.

## 2. Vehicle Modeling

This section is dedicated to the vehicle description, the flight dynamics model, and sensor and actuation models.

### 2.1. Definition of the Vehicle

The proposed GNC approach is applied to an aerial vehicle which features a maneuvering system composed of four canard surfaces, which is roll-decoupled from the main body of the vehicle. The motivation for this aerodynamic configuration is deeply explained in [1]. Note a canard is a small winglike surface attached to an aircraft forward of the main wing to provide extra stability or control, usually replacing the tail. Here, canards are decoupled 2 by 2, to produce control force and its related torque (see [1] for more details on this).

Table 1 shows some characteristic parameters of the vehicle, including thrust typical parameter values, vehicle and fuel mass, inertia, and aerodynamic parameters. These parameters are obtained from fluid dynamics numerical simulations, experimental measurements, and wind tunnel verification (see [1] for clarifications). Note that, to keep continuity and derivability on aerodynamic coefficients and thrust, a cubic splines based interpolation method has been employed at intermediate points. According to the shown moments of inertia, the vehicle features planes of symmetry.

**Table 1.** Aerial vehicle parameters.

| Parameter | Maximum Thrust | | | Initial Mass | |
|---|---|---|---|---|---|
| Value | 29,160.00 N | | | 62.40 kg | |
| Parameter | Fuel mass | | $I_{x0}$ | | $I_{y0}$ |
| Value | 21.00 kg | | 0.19 kg m$^2$ | | 18.85 kg m$^2$ |
| M | $C_{D_0}$ | $C_{D_{\alpha 2}}$ | $C_{L_\alpha}$ | $C_{L_{\alpha 3}}$ | $C_{mf}$ | $C_{Nq}$ |
| 0.00 | 0.27 | 10.74 | 8.01 | 19.82 | −0.59 | 50.81 |
| 0.40 | 0.25 | 10.88 | 8.17 | 19.55 | −0.64 | 53.25 |
| 0.60 | 0.24 | 11.10 | 8.43 | 19.12 | −0.70 | 57.43 |
| 0.70 | 0.24 | 11.24 | 8.60 | 18.83 | −0.72 | 60.31 |
| 0.80 | 0.23 | 11.40 | 8.79 | 18.49 | −0.75 | 63.80 |
| 0.90 | 0.23 | 11.45 | 8.98 | 17.28 | −0.78 | 67.93 |
| 1.00 | 0.41 | 15.12 | 8.93 | 44.05 | −0.81 | 71.38 |
| M | $C_{M_\alpha}$ | $C_{M_{\alpha 3}}$ | $C_{M_q}$ | $C_{mm}$ | $C_{spin}$ | $C_{N\alpha_w}$ |
| 0.00 | −35.58 | −16.65 | −225.73 | 3.02 | −0.04 | 0.00 |
| 0.40 | −35.72 | −18.09 | −232.75 | 3.29 | −0.04 | 0.42 |
| 0.60 | −36.00 | −20.39 | −245.32 | 3.57 | −0.04 | 0.43 |
| 0.70 | −36.21 | −21.82 | −254.10 | 3.71 | −0.03 | 0.44 |
| 0.80 | −36.51 | −23.39 | −264.85 | 3.84 | −0.03 | 0.44 |
| 0.90 | −36.57 | −18.48 | −276.57 | 3.98 | −0.03 | 0.45 |
| 1.00 | −35.99 | 15.39 | −287.82 | 4.12 | −0.03 | 0.45 |

### 2.2. Equations of Flight Mechanics

To construct the equations of flight, three reference frames are defined to project forces and moments: NED axes, working axes and body axes. NED axes, which are ground axes, are depicted by sub index $NED$. They are defined by $x_{NED}$ pointing north, $z_{NED}$ orthogonal to $x_{NED}$ and pointing nadir, and $y_{NED}$ yielding a clockwise trihedron. Working axes are represented by sub index $w$. They are given by $x_w$ pointing to the destination point, $y_w$ orthogonal to $x_w$ and pointing apex, and $z_w$ forming a clockwise trihedron. $AZ_0$ is the initial azimuth, i.e., the azimuth between $x_e$ and $x_w$. Body axes are depicted by sub index $b$. $x_b$ pointing forward and contained in the plane of symmetry of the vehicle, $z_b$ orthogonal to $x_b$ pointing down and contained in the plane of symmetry of the vehicle, and $y_b$ shaping a clockwise trihedron. The origin of body axes is located at the gravity center of the vehicle

and they are rigid coupled to the roll-decoupled control device. Figure 1 shows the previously defined axes systems.

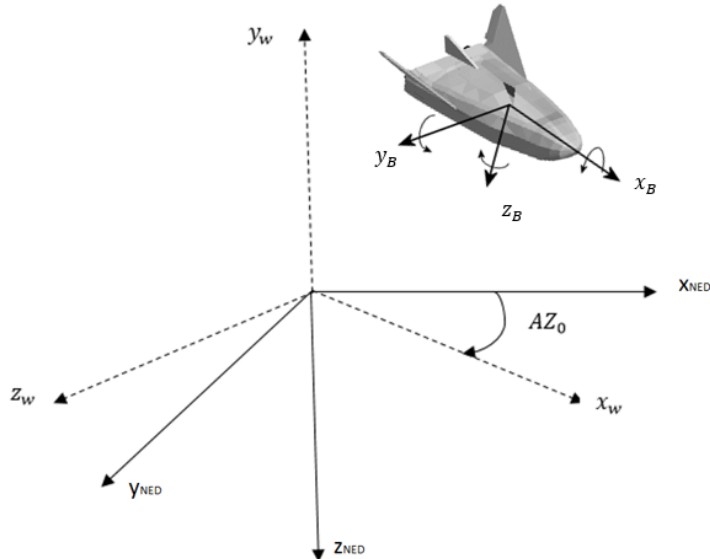

**Figure 1.** The three employed reference frames.

Next, flight dynamics equations are described. Note that these equations are compliant with the standard convention in [27]. Because the vehicle is assumed to be rigid, classical mechanics theory is employed. The Newton–Euler equations describe the combined translational and rotational dynamics of a rigid body. These laws, which are given by six equations, relate the motion of the center of gravity of a rigid body with the sum of forces and moments acting on the rigid body.

$$
\begin{bmatrix} \overrightarrow{F_{ext}} \\ \overrightarrow{M_{ext}} \end{bmatrix} = \begin{bmatrix} \overrightarrow{L} + \overrightarrow{D} + \overrightarrow{P} + \overrightarrow{M} + \overrightarrow{T} + \overrightarrow{W} + \overrightarrow{C} \\ \overrightarrow{P_M} + \overrightarrow{O} + \overrightarrow{M_M} + \overrightarrow{S} \end{bmatrix},
\tag{1}
$$

Equation (1) shows total external forces and moments acting on the vehicle: $\overrightarrow{L}$ is the lift force, $\overrightarrow{D}$ is the drag force, $\overrightarrow{P}$ is the pitch damping force, $\overrightarrow{M}$ is the Magnus force, $\overrightarrow{T}$ is the thrust force, $\overrightarrow{W}$ is the weight force, $\overrightarrow{C}$ is the Coriolis force, $\overrightarrow{P_M}$ the is pitch damping moment, $\overrightarrow{O}$ is the overturn moment, $\overrightarrow{M_M}$ is the Magnus moment, and $\overrightarrow{S}$ is the spin damping moment.

$$
\begin{bmatrix} \overrightarrow{L} \\ \overrightarrow{D} \\ \overrightarrow{P} \\ \overrightarrow{M} \\ \overrightarrow{T} \\ \overrightarrow{W} \\ \overrightarrow{C} \end{bmatrix} = \begin{bmatrix} -\frac{\pi}{8}d^2\rho \begin{bmatrix} \left(C_{L_\alpha}(M)\cdot\alpha + C_{L_{\alpha 3}}(M)\alpha^2\right)\left(\|\overrightarrow{v_w}\|^2\overrightarrow{x_w} - \left(\overrightarrow{x_w}\cdot\overrightarrow{v_w}\right)\overrightarrow{v_w}\right) \\ \left(C_{D_0}(M) + C_{D_{\alpha 2}}(M)\alpha^2\right)\|\overrightarrow{v_w}\|\overrightarrow{v_w} \\ -d\frac{C_{Nq}(M)}{I_y}\|\overrightarrow{v_w}\|^2\left(\overrightarrow{L_w}\times\overrightarrow{x_w}\right) \\ d\frac{C_{mf}(M)}{I_x}\left(\overrightarrow{L_w}\cdot\overrightarrow{x_w}\right)\left(\overrightarrow{x_w}\times\overrightarrow{v_w}\right) \\ T(t)\overrightarrow{x_w} \\ m\overrightarrow{g_w} \\ -2m\overrightarrow{\Omega}\times\overrightarrow{v_w} \end{bmatrix} \end{bmatrix}
\tag{2}
$$

For the computational experiments in this paper, the external forces in working axes are shown in expression (2), where $C_{L_\alpha}$ is the lift force linear coefficient, $C_{L_{\alpha 3}}$ is the lift force cubic coefficient, $\alpha$ is the total angle of attack, $C_{D_0}$ is the drag force linear coefficient, $C_{D_{\alpha 2}}$ is the drag force square coefficient, $\overrightarrow{L_w}$ is the vehicle angular momentum in working axes, $I_x$ *and* $I_y$ are the vehicle inertia

moments in body axes, $C_{Nq}$ is the pitch damping force coefficient, $C_{mf}$ is the Magnus force coefficient, $\overrightarrow{x_w}$ is vehicle pointing vector in working axes, $\overrightarrow{g_w}$ is the gravity vector in working axes, $\overrightarrow{\Omega}$ is earth's angular speed vector, $\overrightarrow{v_w}$ is vehicle velocity in working axes, $d$ is the reference surface of the vehicle, $\rho$ is the air density, and $m$ is the mass of the vehicle. Note that, $\overrightarrow{x_w}$ is the unitary vector of the $x_w$ axis, and $\overrightarrow{g_w}$ is the gravity vector in working axes. Be aware they are nonlinear functions of the variables describing the movement of the vehicle, such as aerodynamic speed, total angle of attack, Mach number, and aerodynamic parameters.

$$
\begin{bmatrix} \overrightarrow{P_M} \\ \overrightarrow{O} \\ \overrightarrow{M_M} \\ \overrightarrow{S} \end{bmatrix} = \frac{\pi}{8} d^3 \rho \begin{bmatrix} \frac{1}{I_y} C_{M_q}(M) \|\overrightarrow{v_w}\| \left( \overrightarrow{L_w} - \left( \overrightarrow{L_w} \cdot \overrightarrow{x_w} \right) \overrightarrow{x_w} \right) \\ \left( C_{M_\alpha}(M) + C_{M_{\alpha 3}}(M)\alpha^2 \right) \|\overrightarrow{v_w}\|^2 \left( \overrightarrow{v_w} \times \overrightarrow{x_w} \right) \\ -\frac{d}{I_x} C_{mm}(M) \left( \left( \overrightarrow{L_w} \cdot \overrightarrow{x_w} \right) \left( (\overrightarrow{v_w} \cdot \overrightarrow{x_w}) \overrightarrow{x_w} \right) - \overrightarrow{v_w} \right) \\ \frac{d}{I_x} C_{spin}(M) \|\overrightarrow{v_w}\| \left( \overrightarrow{L_w} \cdot \overrightarrow{x_w} \right) \overrightarrow{x_w} \end{bmatrix}, \tag{3}
$$

Similarly, the equations in (3) show the mathematical expressions for the moments, including overturning, pitch damping, Magnus, spin damping, and variables and parameters. Here, $C_{M_q}$ is the pitch damping moment coefficient, $C_{M_\alpha}$ is the overturning moment linear coefficient, $C_{M_{\alpha 3}}$ is the overturning moment cubic coefficient, $C_{mm}$ is the Magnus moment coefficient and $C_{spin}$ is the spin damping moment coefficient.

Control forces ($\overrightarrow{CF}$) and moments ($\overrightarrow{CM}$) are obtained from the maneuvering system, which is composed of four canard surfaces. Therefore, the contribution of each of them is summed to obtain the total control forces and moments.

$$
\begin{bmatrix} \overrightarrow{CF} \\ \overrightarrow{CM} \end{bmatrix} = \sum_{i=1}^{i=4} \begin{bmatrix} \frac{1}{8} d^2 \rho \pi \|\overrightarrow{v_b}\|^2 (C_{N\alpha_w}(M)\delta_i)\overrightarrow{n_{ci}} \\ \frac{1}{8} d^3 \rho \pi \|\overrightarrow{v_b}\|^2 (C_{M\alpha_w}(M)\delta_i)(\overrightarrow{x_b} \times \overrightarrow{n_{ci}}) \end{bmatrix} \tag{4}
$$

The expressions in (4) show the mathematical functions for control forces and moments, where $C_{N\alpha_w}$ and $C_{M\alpha_w}$ are the force and moments coefficients of the canard surface respectively, $\overrightarrow{n_{ci}}$ is the normal vector of each canard, and $\delta_i$ is the deflection angle of the canard surface. Here, $\overrightarrow{v_b}$ is vehicle velocity in body axes.

$$
\begin{bmatrix} \overrightarrow{CF} + \overrightarrow{F_{ext}} \\ \overrightarrow{CM} + \overrightarrow{M_{ext}} \end{bmatrix} = \begin{bmatrix} \frac{dm\overrightarrow{v_b}}{dt} + \overrightarrow{\omega_b} \times m\overrightarrow{v_b} \\ \frac{d\overrightarrow{L_b}}{dt} + \overrightarrow{\omega_b} \times \overrightarrow{L_b} \end{bmatrix} \tag{5}
$$

As stated before, a Newton-Euler approach is used to formulate the equations of motion of the aerial vehicle. These equations are in (5). Note that the body-fixed coordinate system (denoted by frame $b$) and the flat-Earth coordinate system (denoted by frame $e$) are related by Euler yaw ($\psi$), pitch ($\theta$), and roll ($\phi$) angles.

In Equations (5), $\overrightarrow{v_b}$ stands for vehicle speed expressed in body axes, $\overrightarrow{\omega_b}$ for angular speed of the vehicle in body axes, and $\overrightarrow{L_b}$ for the angular momentum also in body axes. Recall that control and external forces and control and external moments must be expressed in body axes also to be employed in Equations (5).

### 2.3. Sensors Models

As exposed in the introduction, this research aims at simplifying navigation systems. Here, this means reducing the need for complex and/or expensive sensors. A gyroscope is a device used for measuring orientation and angular velocity, and it is widely employed in navigation systems. However, their precision downgrades for high-dynamics aerial platforms meanwhile costs increase if performance is to be maintained. Therefore, the objective is to avoid them by fusing information from

GNSS sensors, accelerometers, and photo-detector signals to improve vehicle navigation performance in terms of accuracy. This section aims at describing the employed models for these sensors.

### 2.3.1. Global Navigation Satellite System (GNSS)

The GNSS sensor is modeled as a random noise and a bias added to the model calculated position. Note that these systems have typical accuracy of 3 m; therefore, the random noise and bias parameters have been adjusted to satisfy that performance. Because it is not the objective of this paper to model such a sensor, the reader is referred to [2,28] for more details on this.

These kind of sensors provide good performance during intermediate phases of flight and are employed to determine the line of sight vector expressed in NED axes. Note that GNSS sensors also provide velocity vector in NED axes. This vector is also modeled as a random noise and a bias of $0.1 \text{ ms}^{-1}$, which resembles real performance of these sensors.

### 2.3.2. Accelerometers

An accelerometer is a device that measures acceleration, i.e., the rate of change of velocity of the vehicle in its own instantaneous reference frame. They are modeled as a random noise and bias of $0.001 \text{ ms}^{-2}$ which resembles real performance of these sensors. Because they provide the acceleration vector expressed in body axes, velocity vector expressed in body triad can be obtained after integration of each of its components along time.

In addition, the magnitudes obtained from accelerometers will be used in the gravity vector estimation approach. As it is explained in the following sections, the velocity vector module is required to estimate it.

### 2.3.3. Semiactive Laser Kit

Laser guidance may be used to guide a vehicle to a target by means of a laser beam. With this method, a laser is kept pointed at the objective and the laser radiation bobs off the objective and is dispersed every which way. At the point when the vehicle is close enough for a portion of the reflected laser energy from the objective to arrive at it, a laser seeker detects which direction this energy is coming from and provides a signal to correct the trajectory towards the source. The device seeking the laser and providing the signal is a semiactive laser kit.

The signal provided by this sensor features the centroid of the laser footprint in the photo-detector of the kit, which is composed of four photodiodes that convert light into an electrical current. To estimate its coordinates, the produced electrical intensities in the photodiodes ($I_1$, $I_2$, $I_3$ and $I_4$) are employed, which rely upon the illuminated area. These coordinates can be determined as $[ln \frac{I_4}{I_2}, ln \frac{I_1}{I_3}]$ [17], and from them, it is possible to obtain the measured radial distance, $r_{quad}$. Notwithstanding, real coordinates differ from those calculated, although the transformation is conformal [17]. To obtain the real radial distance, $r_c$, the following mathematical functional relationship may be employed: $r_c = f(r_{quad})$. Then, cubic splines based interpolation is applied to obtain a continuous relationship. Equations (6) are utilized to estimate $x_c$ and $y_c$ (see [17] fore more details on this), which are the real spot center coordinates:

$$\begin{bmatrix} x_c \\ y_c \end{bmatrix} = R_{quad} \cdot \frac{r_c}{r_{quad}} \begin{bmatrix} ln \frac{I_4}{I_2} \\ ln \frac{I_1}{I_3} \end{bmatrix}, \tag{6}$$

where the physical radius of the photo-detector of the kit is given by $R_{quad}$. Consequently, the line of sight vector projected in body axes may be calculated from $x_c$ and $y_c$ and also from the distance of the photo-detector of the kit to the center of mass of the vehicle.

Note that the signal of this sensor is only available during the terminal phase of the flight. However, it is during final stages of flight when errors of 3 m in positioning target and vehicle induces enormous errors. In this way, an accurate terminal guidance sensor, for example, a semiactive laser kit,

is suggested for these last flight stages. This semiactive laser sensor, in combination with GNSS and accelerometers, will provide an accurate determination of the line of sight, especially in the terminal phase. For that purpose, the signals of these sensors' must be hybridized.

Next, GNC algorithms are presented. At their kernel, neural networks are implemented to determine gravity vector in two reference frames in order to determine vehicle attitude. In addition, hybridization algorithms are applied to sensors' signals to improve precision.

## 3. Guidance, Navigation and Control (GNC) Algorithm Definition

This section details the proposed navigation, guidance and control algorithms. A scheme of the overall process is depicted in Figure 2. The navigation function determines the position and attitude of the vehicle by means of the information sensed by the sensors. The position is determined through the integration of the signals provided by the accelorometers and the hybridization of the signal from a GNSS device. The determination of attitude is the core of the research in this paper. From the information provided by the accelerometers and the GNSS, the neural network determines the gravity vector in two different axes systems. Then Euler angles are devised from a triad algorithm. The guidance function compares the information from the navigation function with a reference and computes a desired action to the control function. The control function processes this desired action and transforms it into a command to the actuators of the plant (i.e., the vehicle), which execute the action. The action taken is again measured by the sensors, which closes the loop.

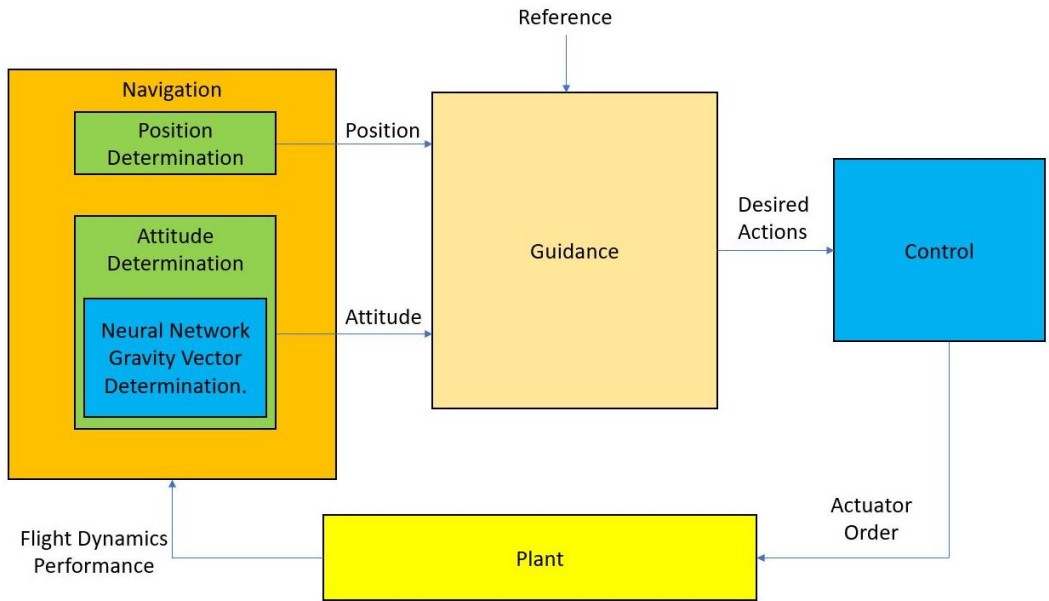

**Figure 2.** Scheme of the GNC process.

### 3.1. Navigation

Navigation refers to the determination during the flight, of the position and attitude of the vehicle, and target position.

Determining the position of the vehicle may consist of integrating accelerometers' measurements to be hybridized with GNSS sensor information. The details of these calculations are out of the scope of this paper, see [1,2,28,29] for details on this approach.

As it was mentioned before, determining attitude involves knowing two or more different vectors in two different reference systems. The velocity vector and the line of sight vectors can be the two needed vectors. If a GNSS sensor device is equipped on the aircraft, velocity vector can be directly obtained from sensor measurements in the NED triad, which can be expressed as shown in (7), where $v_{x_{NED}}$, $v_{y_{NED}}$ and $v_{z_{NED}}$ are the components of this velocity in NED axes.

$$\overrightarrow{v_{NED}} = [v_{x_{NED}}, v_{y_{NED}}, v_{z_{NED}}]^T \tag{7}$$

In parallel, the same velocity vector can also be calculated in body triad from a set of accelerometers, one on each of the axes. Integrating each of their measures along time, the velocity vector is obtained as shown in (8). Here, $a_{x_B}, a_{y_B}$ and $a_{z_B}$ are the components of the acceleration in body axes as measured by the accelerometers and $\overrightarrow{\omega_b}$ is the estimated angular speed expressed in body axes. Note that, at this point, $\overrightarrow{\omega_b}$ is unknown, and the algorithm for estimating it will be shown in the following sections.

$$\overrightarrow{v_B} = \int \left\{ [a_{x_B}, a_{y_B}, a_{z_B}]^T + \overrightarrow{\omega_b} \times \overrightarrow{v_B} \right\} dt \tag{8}$$

Similarly, the line of sight vector must be obtained in NED and body reference systems, $\overrightarrow{LOS_{NED}}$ and $\overrightarrow{LOS_B}$, respectively. $\overrightarrow{LOS_{NED}}$ can be easily obtained from GNSS sensor information. However, the semiactive laser kit is needed to derive $\overrightarrow{LOS_B}$, and this sensor signal is not available until the vehicle is close enough to the target. This means another vector is needed to successfully estimate the attitude of the vehicle during all the phases of flight.

The gravity vector poses as a natural candidate for such a challenge. Notice that determining the gravity vector in NED triad is straightforward. It is always parallel to $\overrightarrow{z_{NED}}$. Its expression is shown in (9), where $g$ is the gravity acceleration, which is a fixed constant in this model (9.81 m/s$^2$). Note that precision may be increased using more sophisticated models, i.e., it can be made variable with longitude, latitude, and altitude.

$$\overrightarrow{g_{NED}} = g[0, 0, 1]^T \tag{9}$$

However, the gravity vector expressed in another reference system, i.e, body axes, is also needed. However, although there are multiple available approaches to obtain it, none of them is simple and/or require additional sensors. For example, it can be estimated determining the constant component of the measured acceleration employing a low pass filter, where Jerk in body axes is calculated by derivation of the acceleration; then, it is integrated to obtain the nonconstant component of acceleration and, by subtracting this nonconstant component from the measured acceleration, gravity vector may be estimated. However, this method is not valid when the aircraft rotates. Another method to obtain the gravity vector is to integrate the mechanization equations [30] to obtain it from the resulting expressions. However, gyros are needed to implement this method. Therefore, the keystone of the presented attitude determination method is determining gravity vector in body axes.

An estimation method for the gravity vector, which is valid for nonrotating and rotating aircraft and which is only based on accelerometers, is presented in the following subsection.

### 3.1.1. Neural Network Based Gravity Vector Estimation

Among the numerous applications that machine learning offers to exemplary and current GNC issues (see [23,31–34]), its potential to precisely estimate the gravity vector from sensor information is one of the main unexplored settings. The utilization of neural networks (NN) to understand the evolution of nonlinear equations has been demonstrated before [35], regardless of uncertainty. Scientifically, this infers NN will learn flight mechanics equations [36] and produce an equal outcome.

The gravity vector estimation method presented here depends on the aerial platform on which it will be employed. This means that the method must be adjusted for the aircraft of interest. Without loss of generality, the estimation method detailed in this section is particularized for a four canard controlled aerial vehicle. However, using the appropriate neural network training, it may be applied to other aircraft.

The estimated gravity vector may be expressed as shown in (10), where its components in body axes are displayed. The point is to recoup a high precision gravity vector by consolidating the estimations from the accelerometers and the potential offered by machine learning.

$$\overrightarrow{\widetilde{g_B}} = [\widetilde{g_{x_B}}, \widetilde{g_{y_B}}, \widetilde{g_{z_B}}]^T \tag{10}$$

In order to prove the suitability of the proposed approach two different methods or strategies are proposed, as they can be visualized in Table 2:

- Method 1: it is based on a neural network which features two-layers with one hundred standard sigmoid hidden neurons and the usual linear output neuron [35]. The input vector is composed of three components constructed from $a_{x_B}$, $a_{y_B}$ and $a_{z_B}$. The outputs of the neural networks are the components of the gravity vector expressed in body axes ($\overrightarrow{\widetilde{g_B}}$).
- Method 2: this method is the same as method 1, but the number of neurons in the hidden layer is 50.

**Table 2.** Neural network schemes for the two different methods or strategies.

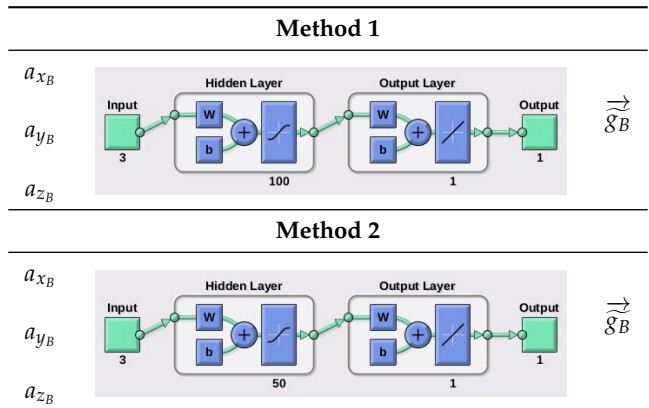

The choice of the number and shape of neurons as well as the amount of training and validation data selected for the presented two strategies is a result of the literature review (specially from [35]) and a performed hyperparametric study. This study provided two points of interest: around 100 neurons and 50 neurons. Increasing the number or neurons or layers translated only into an increase of computation time for a limited improvement in terms of error of approximation. A further and detailed hyperparametric study will be performed in future work to precisely determine the optimal working point but is not the objective of this research to formalize this statement. The preliminary results of this hyperparametric study suggest that there is a limit number of neurons in the intermediate layer (estimated at about 100 neurons), and over this limit, results do not get improved.

Then, neural networks are trained replicating the flight dynamics problem. Two examples of the available $3 \cdot 10^8$ rows of data, which are obtained from 12,000 simulations, are showed in Table 3.

**Table 3.** Neural network input and target values.

| Accelerometer Inputs | | | Target | | |
|---|---|---|---|---|---|
| $a_{x_B}$ | $a_{y_B}$ | $a_{z_B}$ | $\widetilde{g_{x_B}}$ | $\widetilde{g_{y_B}}$ | $\widetilde{g_{z_B}}$ |
| −17.68 | −0.004 | 9.761 | −0.614 | −0.002 | 9.791 |
| −6.019 | 5.605 | 0.1891 | 5.838 | −0.093 | 7.883 |
| ... | ... | ... | ... | ... | ... |

Regarding the training process, three different backpropagation algorithms are employed: Scaled Conjugate Gradient (SCG) [37], Bayesian regularization (BR) [38,39], and Levenberg-Marquardt backpropagation (LM) [40,41]. The choice of these algorithms is a result of literature study. The percentage of data employed in this training is 70%. As it is common practice, a representative amount of sensor data and its corresponding gravity vector are left aside for validation purposes. In this case, a 15% of the available data corresponds to validation data. Note that the total amount of methods and training algorithms provide six different combinations which are analyzed next.

The performance of each of the six approaches can be quantified by means of the Mean Squared Error (MSE) and the Regression (R) parameter values. The MSE is the average squared difference between outputs and targets. Lower value means lower error. Zero means no error. R values measure the correlation between outputs and targets. An R value of 1 means a close relationship and 0 a random relationship. Other kind of error indicators (such as Mean Average Error, MAE) may also be used to monitor and validate the training to avoid overfitting.

For each of the the training processes, a maximum number of 1000 iterations has been established. As it is common practice in the field, classified by epochs. For the LM and SCG algorithms, training automatically stops when generalization stops improving, as indicated by an increase in the MSE of the validation samples. In the case of the BR algorithm, training stops according to adaptive weight minimization (regularization). In both cases the MAE is controlled as usual to avoid overfitting.

In addition, the trained NN is tested with the independent data (15% of the collected data), and MSE and R values are also calculated to validate the presented strategies.

Table 4 summarizes the obtained results for the training, validation and tests. It shows the values for the MSE and the R parameters. In the first column, the "Set" of data is defined, i.e., train (70%), validation (15%), or test (15%) data. The second column displays the employed training algorithm. The third and forth columns present the MSE and R values for the methods 1 and 2 showed in Table 2.

**Table 4.** MSE and R values for neural network based gravity vector estimator.

| Set | Alg. | Method 1 | | Method 2 | |
|---|---|---|---|---|---|
| | | MSE | R | MSE | R |
| Train | SCG | $5.81 \times 10^{-3}$ | $7.22 \times 10^{-1}$ | $5.82 \times 10^{-3}$ | $7.24 \times 10^{-1}$ |
| Validation | SCG | $5.89 \times 10^{-3}$ | $7.21 \times 10^{-1}$ | $5.98 \times 10^{-3}$ | $7.25 \times 10^{-1}$ |
| Test | SCG | $5.94 \times 10^{-3}$ | $7.18 \times 10^{-1}$ | $5.85 \times 10^{-3}$ | $7.21 \times 10^{-1}$ |
| Train | BR | $6.95 \times 10^{-5}$ | $9.98 \times 10^{-1}$ | $9.83 \times 10^{-4}$ | $9.55 \times 10^{-1}$ |
| Validation | BR | $6.92 \times 10^{-5}$ | $9.98 \times 10^{-1}$ | $9.80 \times 10^{-4}$ | $9.54 \times 10^{-1}$ |
| Test | BR | $6.95 \times 10^{-5}$ | $9.97 \times 10^{-1}$ | $9.81 \times 10^{-4}$ | $9.56 \times 10^{-1}$ |
| Train | LM | $5.63 \times 10^{-5}$ | $9.98 \times 10^{-1}$ | $7.31 \times 10^{-4}$ | $9.60 \times 10^{-1}$ |
| Validation | LM | $5.61 \times 10^{-5}$ | $9.98 \times 10^{-1}$ | $7.21 \times 10^{-4}$ | $9.72 \times 10^{-1}$ |
| Test | LM | $5.70 \times 10^{-5}$ | $9.97 \times 10^{-1}$ | $7.32 \times 10^{-4}$ | $9.65 \times 10^{-1}$ |

Analyzing the results in Table 4, it may be concluded that the best results are obtained for the combination of Method 1 and LM algorithm, which yields a MSE value of $5.63 \cdot 10^{-5}$ and a Regression value of 0.998. Additionally, the combinations of Method 2 and LM and BR algorithms also result in acceptable values for MSE and R values, they are of the same order of magnitude. Consequently, we may conclude that Methods 1 and 2 provide good results when the LM and BR training algorithms are used. Nevertheless, the SCG training algorithm is not appropriate for this application, as the best results for this algorithm are 2 orders of magnitude worse as compared to the the rest of the algorithms.

Next, the attitude determination algorithm is presented. It is based on the estimated gravity vector by the neural networks (NN). In addition, note that, because there is information regarding

two additional vectors during terminal flight, i.e., the speed vector and the line of sight vector, a hybridization approach is also presented to improve performance.

### 3.1.2. Attitude Determination Algorithm

Attitude determination can be determined by solving a classical Wahba's problem [42]. An orthonormal base must be defined in both axes systems, B and NED. This orthonormal base is defined for intermediate phases of flight, when signal of the photo-detector is not available and for the terminal phase of flight, when it is available, by unitary vectors $\vec{i}, \vec{j}$ and $\vec{k}$ expressed in both bases. For the intermediate phases, these unitary vectors are calculated using the speed vector and the gravity vector. Furthermore, for the terminal flight, the line of sight vector and the gravity vector are to be employed. For the intermediate phases, $fl$, the unitary vectors are defined by expressions (11) and (12).

$$\overrightarrow{i_{NED_{fl}}} = \frac{\overrightarrow{v_{NED}}}{\left\|\overrightarrow{v_{NED}}\right\|}, \quad \overrightarrow{j_{NED_{fl}}} = \frac{\overrightarrow{v_{NED}} \times \overrightarrow{g_{NED}}}{\left\|\overrightarrow{v_{NED}} \times \overrightarrow{g_{NED}}\right\|}, \quad \overrightarrow{k_{NED_{fl}}} = \frac{\overrightarrow{i_{NED_{fl}}} \times \overrightarrow{j_{NED_{fl}}}}{\left\|\overrightarrow{i_{NED_{fl}}} \times \overrightarrow{j_{NED_{fl}}}\right\|} \tag{11}$$

$$\overrightarrow{i_{B_{fl}}} = \frac{\overrightarrow{v_B}}{\left\|\overrightarrow{v_B}\right\|}, \quad \overrightarrow{j_{B_{fl}}} = \frac{\overrightarrow{v_B} \times \overrightarrow{g_B}}{\left\|\overrightarrow{v_B} \times \overrightarrow{g_B}\right\|}, \quad \overrightarrow{k_{B_{fl}}} = \frac{\overrightarrow{i_{B_{fl}}} \times \overrightarrow{j_{B_{fl}}}}{\left\|\overrightarrow{i_{B_{fl}}} \times \overrightarrow{j_{B_{fl}}}\right\|} \tag{12}$$

During the terminal guidance phase, $tf$, when the photo-detector is receiving information, a new set of unitary vectors is obtained by Equations (13) and (14).

$$\overrightarrow{i_{NED_{tf}}} = \frac{\overrightarrow{LOS_{NED}}}{\left\|\overrightarrow{LOS_{NED}}\right\|}, \quad \overrightarrow{j_{NED_{tf}}} = \frac{\overrightarrow{LOS_{NED}} \times \overrightarrow{g_{NED}}}{\left\|\overrightarrow{LOS_{NED}} \times \overrightarrow{g_{NED}}\right\|}, \quad \overrightarrow{k_{NED_{tf}}} = \frac{\overrightarrow{i_{NED_{tf}}} \times \overrightarrow{j_{NED_{tf}}}}{\left\|\overrightarrow{i_{NED_{tf}}} \times \overrightarrow{j_{NED_{tf}}}\right\|} \tag{13}$$

$$\overrightarrow{i_{B_{tf}}} = \frac{\overrightarrow{LOS_B}}{\left\|\overrightarrow{LOS_B}\right\|}, \quad \overrightarrow{j_{B_{tf}}} = \frac{\overrightarrow{LOS_B} \times \overrightarrow{g_B}}{\left\|\overrightarrow{LOS_B} \times \overrightarrow{g_B}\right\|}, \quad \overrightarrow{k_{B_{tf}}} = \frac{\overrightarrow{i_{B_{tf}}} \times \overrightarrow{j_{B_{tf}}}}{\left\|\overrightarrow{i_{B_{tf}}} \times \overrightarrow{j_{B_{tf}}}\right\|} \tag{14}$$

Note that to determine the attitude of a vehicle with respect to a reference frame, the direction cosine matrix (DCM) must be determined. It represents the attitude of the body frame (B) relative to the reference frame (NED). It is specified by a $3 \times 3$ rotation matrix, where the columns represent unit vectors in the body axes projected along the reference axes. Therefore, the expression to determine the DCM is as shown in Equation (15), where $\left[\overrightarrow{i_{B_i}}, \overrightarrow{j_{B_i}}, \overrightarrow{k_{B_i}}\right]$ is a $3 \times 3$ square matrix composed of orthonormal vectors in body triad, $\left[\overrightarrow{i_{NED_i}}, \overrightarrow{j_{NED_i}}, \overrightarrow{k_{NED_i}}\right]$ expresses the same concept in NED triad, and $DCM_{B,NED_i}$ is the director cosine matrix that transforms NED triad into body triad. Notice that depending on the phase of flight, i.e., intermediate ($fl$) or terminal ($tf$), the matrix may be calculated with different inputs.

$$\left[\overrightarrow{i_{B_i}}, \overrightarrow{j_{B_i}}, \overrightarrow{k_{B_i}}\right] = DCM_{B,NED_i} \left[\overrightarrow{i_{NED_i}}, \overrightarrow{j_{NED_i}}, \overrightarrow{k_{NED_i}}\right] \forall i \in \{tf, fl\} \tag{15}$$

The DCM matrix can be solved from Equation (15) as it is shown in Equation (16). Employing an orthonormal base simplifies the calculation of the inverse matrix as it is the transposed matrix.

$$DCM_{B,NED_i} = \left[\overrightarrow{i_{B_i}}, \overrightarrow{j_{B_i}}, \overrightarrow{k_{B_i}}\right] \left[\overrightarrow{i_{NED_i}}, \overrightarrow{j_{NED_i}}, \overrightarrow{k_{NED_i}}\right]^T \forall i \in \{tf, fl\} \tag{16}$$

After obtaining the two different director cosine matrices, which will be essentially similar matrices, the rotation is characterized. The most suitable method to express this rotation is through quaternions, as they avoid any possible singularities on the poles of rotation. It is widely known that quaternions themselves are enough to express rotations without singularities, but it is also known that conceptually they are difficult to be visualized. An easier manner to define these rotations is by means of Euler angles. Concretely, the most common aeronautical rotation is defined by roll ($\phi_i$), pitch ($\theta_i$), and yaw ($\psi_i$) angles for $i \in \{tf, fl\}$. A method to obtain a quaternion solution is explained

in [2]. Note that two different values for each quaternion are obtained, $i \in \{tf, fl\}$. This fact requires an hybridization between them in order to only obtain one value for each quaternion.

Hybridization Algorithm

The Euler angles (or their corresponding quaternions) values obtained for $i \in \{tf, fl\}$ are hybridized applying the recursive algorithm described in (17) and (18):

$$
\left\{\overrightarrow{Eul}\right\}\Big|_n = \begin{cases} \left\{\overrightarrow{Eul}\right\}\Big|_{n-1} + \kappa\big|_n \left[\left\{\overrightarrow{Eul_{fl}}\right\}\Big|_n - \left\{\overrightarrow{Eul}\right\}\Big|_{n-1}\right] & \text{if } \nexists \left\{\overrightarrow{Eul_{tf}}\right\}\Big|_n \\[2mm] \left\{\overrightarrow{Eul}\right\}\Big|_{n-1} + \kappa\big|_n \left[\left\{\overrightarrow{Eul_{tf}}\right\}\Big|_n - \left\{\overrightarrow{Eul}\right\}\Big|_{n-1}\right] & \text{if } \exists \left\{\overrightarrow{Eul_{tf}}\right\}\Big|_n \end{cases} \tag{17}
$$

$$
\kappa\big|_n = \Gamma \cdot [\Gamma + \Lambda]^{-1}, \tag{18}
$$

where $\overrightarrow{Eul}$ are the Euler angles $(\phi, \theta, \psi)$, and $\Gamma$ and $\Lambda$ are the error covariance matrices for $i = tf$ and for $i = fl$ measurements, which are set to $1.3 \cdot 10^{-6}$ and $0.95 \cdot 10^{-3}$, respectively. Those values were determined empirically.

The Euler angles obtained in (17) may be used to characterize rotations and angular speeds in navigation, guidance, and control algorithms. This basically means that $\overrightarrow{\omega_b}$ is now known. Furthermore, from these Euler angles, the hybridized director cosine matrix ($DCM_{B,NED}$) may be calculated.

*3.2. Guidance Law*

Guidance is given in two stages. The first comprises of a constant angle glide trajectory, while the second one is based on a modified proportional law.

3.2.1. Constant Angle Glide Trajectory

Equation (19) proposes a law which is chosen to increase range. It adjusts the longitudinal axis of the vehicle ($x_b$) with a vertical flight plane, orthogonal to ground, parallel to the line joining the gravity center of the vehicle and the destination target and containing the gravity center of the vehicle. The line of sight is expressed in working axes is given by vector $\overrightarrow{LOS_w} = [LOS_{1_w}, LOS_{2_w}, LOS_{3_w}]$. $x_b$ in working axes is represented by vector $\overrightarrow{x_{b_w}} = [x_{b1_w}, x_{b2_w}, x_{b3_w}]$. Consequently, the lateral correction to be applied ($\psi_{dem}$) is calculated by the first component of Equation (19), while the correction in the vertical plane ($\theta_{dem}$) with respect to a constant glide angle trajectory given by $C_1$ [1] is given by the second component. Note that guidance effectively starts after apogee, which is determined by the pitch angle ($\theta$) and after fuel burn time.

$$
\begin{bmatrix} \psi_{dem} \\ \theta_{dem} \end{bmatrix} = \begin{cases} \begin{bmatrix} \left(atan\,\dfrac{LOS_{3_w}}{LOS_{1_w}} - atan\,\dfrac{x_{b3_w}}{x_{b1_w}}\right) \\ C_1 \end{bmatrix} & \text{if } t > 5 \text{ and } \theta \leq 0 \\[4mm] \begin{bmatrix} 0 \\ 0 \end{bmatrix} & \text{else} \end{cases} \tag{19}
$$

3.2.2. Modified Proportional Law

The guidance for the terminal phase of flight is formulated as a modified proportional law ruled by expression (20). Equation (21) calculates time to target, $t_{go}$. Guidance is activated only when the vertical coordinate of the line of sight vector is greater than a given constant ($C_2$) [1].

$$
\begin{bmatrix} \psi_{dem} \\ \theta_{dem} \end{bmatrix} = \begin{cases} \dfrac{\overrightarrow{LOS_w} - \overrightarrow{v_w}t_{go}}{t_{go}^2} \cdot \begin{bmatrix} \overrightarrow{k_w} \\ -\overrightarrow{i_w} \end{bmatrix} & \text{if } atan\,\dfrac{LOS_{3_{NED}}}{\sqrt{LOS_{1_{NED}}^2 + LOS_{2_{NED}}^2}} \leq C_2 \\[4mm] \begin{bmatrix} 0 \\ 0 \end{bmatrix} & \text{else} \end{cases} \tag{20}
$$

$$t_{go} = max \begin{bmatrix} \frac{1}{g}\left(\vec{v_w}\cdot\vec{j_w} + \sqrt{(\vec{v_w}\cdot\vec{j_w})^2 + 2gLOS_{2_w}}\right) \\ \frac{1}{g}\left(\vec{v_w}\cdot\vec{j_w} - \sqrt{(\vec{v_w}\cdot\vec{j_w})^2 + 2gLOS_{2_w}}\right) \end{bmatrix} \tag{21}$$

Here, $\vec{i_w}, \vec{j_w}$, and $\vec{k_w}$ represent the orthonormal basis of the working axes, and $\overrightarrow{LOS_w} = [LOS_{1_w}, LOS_{2_w}, LOS_{3_w}]^T$ and $\overrightarrow{LOS_{NED}} = [LOS_{1_{NED}}, LOS_{2_{NED}}, LOS_{3_{NED}}]^T$ are the vectors of the line of sight and its components in working and NED axes, respectively.

### 3.3. Control System

The control law presented in [29] is employed in the current research. Two control conditions are presented in the actuation device: the modulus and the angle for the control force. Control is computed by a double loop feedback system. The inner loop aims at augmenting the stability of the vehicle. Equation (22) characterizes modulus ($\tau_c$) and angle ($\phi_c$) of the control force. Its inputs are pitch ($\theta_{dem}$) and yaw ($\psi_{dem}$) errors. $K_i$, $K_d$ and $K_p$ are the integral, derivative and proportional constants of the controller, $K_{mod}$ is a constant to adjust force module and $L1$ and $L2$ are experimental gains. The procedure to decide these constant values, which appear in Table 5, is clarified in [1]. Note that the acceleration, without accounting for gravity, of the vehicle in body axes is defined by $[acc_{xb}, acc_{yb}, acc_{zb}]$. Euler angles as introduced before are represented by $[\phi, \theta, \psi]$.

$$\begin{bmatrix} \phi_c \\ \tau_c \end{bmatrix} = \begin{bmatrix} K_p\,(E1 - E2) + K_i \int (E1 - E2)\,dt + K_d \frac{d}{dt}(E1 - E2) + E1 \\ K_{mod}\sqrt{(\theta_{dem} - L_1\theta)^2 + (L_2(\psi_{dem} - L_1\psi))^2} \end{bmatrix}$$

$$\text{where } \begin{cases} E1 = atan\,\frac{\theta_{dem} - L_1\theta}{L_2(\psi_{dem} - L_1\psi)} \\ E2 = atan\,\frac{acc_{zb}}{acc_{yb}} \end{cases} \tag{22}$$

**Table 5.** Values for the constants of the control systems for each flight phase.

| Phase of Flight | $C_1$ | $C_2$ | $K_i$ | $K_p$ | $K_d$ | $K_{mod}$ | $L_1$ | $L_2$ |
|---|---|---|---|---|---|---|---|---|
| Intermediate phases | −7.5 deg | −21 deg | 0 | 0.5 | 0 | 0.08 | 0.01 | 100 |
| Terminal phase | −7.5 deg | −21 deg | 1 | 0.25 | 0.05 | 0.08 | 0.01 | 1 |

Summarizing, the control law works as follows. The controller determines the required pointing angle of the aerodynamic force. This is calculated obtaining the arc-tangent of the quotient of the pitch and yaw error, which provides an angle, in the $y_b - z_b$ plane, at which the aerodynamic force should point to reach the objective. However, due to gyroscopic effects, the response of the vehicle is hard to govern, i.e., pointing the control force upwards may not make the vehicle to react upwards. Consequently, knowing the acceleration of the vehicle, without accounting for gravity, is a must. Similarly, the difference between $\phi_c$ and the angle the projection of the aerodynamic force in the $y_b - z_b$ plane forms with $y_b$ needs to be determined [1].

The aforementioned parameters of control are transformed into canard surface deflections, i.e., $\delta_1, \delta_2, \delta_3$ and $\delta_4$, which are ruled by two different actuators, as it is shown in Equation (23).

$$\begin{bmatrix} \delta_1 \\ \delta_2 \\ \delta_3 \\ \delta_4 \end{bmatrix} = \tau_c \begin{bmatrix} sin\phi_c \\ cos\phi_c \\ sin\phi_c \\ cos\phi_c \end{bmatrix} \tag{23}$$

## 4. Numerical Simulations

The described nonlinear dynamics are integrated forward in time utilizing a fixed time step Runge–Kutta method of fourth grade to get a single flight path. [1] shows the validation of this modeling and solving approach for aerial platforms. To demonstrate the precision of the novel methodology introduced in this research, which is based on neural networks, the obtained results are compared to the obtained outcomes in [29]. The methodology in [29] features a Kalman based hybridization [43,44] of GNSS, IMU and semiactive laser quadrant photo-detector sensors. To integrate the equations of motion and their interactions with GNC system and environment, MATLAB/Simulink R2020a on a personal computer with a processor of 2.8 Ghz and 32 GB RAM is used.

The remainder of this section is separated in two subsections. The first one presents the noncontrolled trajectories to which the developed navigation, guidance, and control algorithms will be applied. The second one performs Monte Carlo simulations of ballistic flights, Kalman hybridization based controlled flights, and neural networks based controlled flights. Moreover, an ideal controller without induced errors in the line of sight is also developed to compare results with ideal results.

### 4.1. Noncontrolled Trajectories

Three nominal trajectories are established to test the developed approach. Each one differs in its initial pitch angle: 20°, 30° and 45°. Destination points are at 18,790 m, 23,007 m, and 26,979 m, respectively. In order to compensate Coriolis and gyroscopic forces, initial lateral correction is set to 0.15°, 0.19° and 0.31°, respectively. Figure 3 shows many of these trajectories in a 3D setting for different settings.

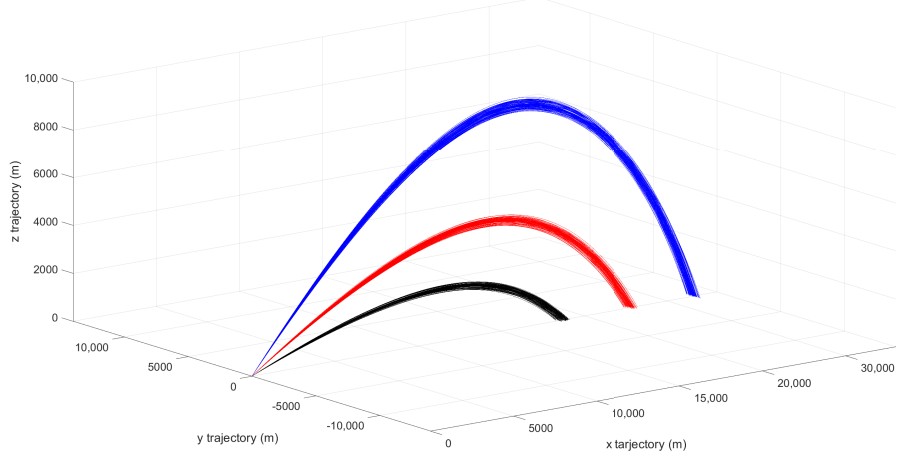

**Figure 3.** Noncontrolled flights for 20° (black), 30° (red) and 45° (blue) initial pitch angles.

### 4.2. Monte Carlo Simulations

Noncontrolled flights have been validated against real data provided by the Spanish air force. Monte Carlo simulations are performed to calculate closed-loop performance over a full range of uncertainty settings, which have been defined with the support of the Spanish air force. These settings model the potential uncertainty that can arise in aspects such as initial conditions, sensor information procurement, weather conditions, and thrust properties. Note that, details on uncertainty models for sensors are given in the previous sections. Table 6 shows mean and standard deviation for the rest of the considered uncertain parameters.

A set of 2000 flights is performed for each of the following approaches: noncontrolled flights, Kalman hybridization based controlled flights, neural network based controlled flights, and ideally controlled flights. Simulations are run for each of the initial pitch angles. Note that the previously used data for neural network training is different from the data employed in the simulations in this section.

**Table 6.** Monte Carlo simulation parameters.

| Parameter (deg) | Initial $\phi$ | Initial Pitch | Wind Speed | Wind Direction | Thrust | Initial Azimuth |
|---|---|---|---|---|---|---|
| Mean | 0° | Nom. | 10 m/s | 0° | T(t) | Nom. |
| Standard Deviation | 20° | 0.01° | 5 m/s | 20° | 10 N | 0.01° |

*4.3. Discussion*

Results for noncontrolled trajectories are shown in Figure 3, which depicts destination point dispersion patterns. This figure shows the ballistic trajectory that the vehicle follows for three different launch angles without using the control system at all, that is, it shows the flight of the system before implementing the improvements provided by the GNC system. The circular error probable (CEP), which is defined as the radius of a circle, centered on the mean, whose boundary is expected to include the landing points of 50% of the flights. The CEP is employed as a quality check parameter at the final step of the simulation as it is a valid reference for any utilized method. Indeed, the lower the CEP is the better the global GNC device is.

Table 7 displays the CEP for noncontrolled flights, Kalman hybridization based controlled flights and ideally controlled flights for each of the initial pitch angles.

**Table 7.** The CEP for noncontrolled flights, Kalman hybridization based controlled flights and ideally controlled flights for 20°, 30° and 45° initial pitch angles.

| Initial Pitch Angle (deg) | Noncontrolled CEP (m) | Kalman Based CEP (m) | Ideal CEP (m) |
|---|---|---|---|
| 20 | 169.34 | 1.28 | 1.18 |
| 30 | 239.37 | 1.18 | 1.06 |
| 45 | 281.59 | 0.98 | 0.83 |

Analyzing Table 7, it may be concluded that the CEP increases with target distance for noncontrolled flights, as expected. However, it almost remains constant for either Kalman based or ideally controlled flights, which means the use of an appropriate GNC device eliminates the correlation between the CEP and the distance to the objective. Recall, the main purpose is to show that these results are reproducible when employing machine learning and when reducing the availability of sensors.

Table 8 shows again the CEP for different trajectories, now as obtained by the novel presented approach. Each row, excluding the first one, which shows the headings of the columns, displays the resulting CEPs for every combination of trajectory and NN training algorithm. The first column in the table identifies the trajectory, the second one the training algorithm, the third one the CEP for the NN architecture in method 1, and the last one the CEP for the NN architecture in method 2. One of the main conclusions drawn from Table 8 is that the results for the SCG training algorithm present an unacceptable big CEP, which means low accuracy when reaching destination. Consequently, this training algorithm should be discarded in this case. Indeed, we recover results equivalent to a defective GNC system, even showing worse performance than ballistic flights. Several tests and a hyperparametric analysis were conducted, and it is observed that these kinds of errors were systematic, concluding that this training algorithm does not match well with the fed data to the NN. However, the results from BR and LM training algorithms show a good behavior throughout the trajectory, which means level of accuracy at destination is high. Diving into the numerical results in Table 8, it can be stated that the presented novel approach, which relaxes sensor requirements, is even able of outperforming the Kalman hybridization based approach. It can also be observed that the results are coherent with the training results depicted in Table 4. For example, the poor MSE and R values for the SCG training algorithm are reflected in the unacceptable GNC device results. Comparing the

obtained CEPs to what it was obtained in [1,22,29], it can be concluded that the results here are of the same order of magnitude and that the NN algorithms are viable for these type of applications.

As a summary, results for ballistic trajectories and comparisons between different approaches are shown in Figure 4. It is composed of four columns and three rows of subfigures. Each row features a different initial pitch angle. The first column of subfigures compares ballistic flights against Kalman hybridization assisted flights, the second one compares Kalman hybridization against neural network hybridization, the third one neural network hybridization against ideal controller, and the last one ballistic flights against neural network hybridization assisted flights. Note that, even with an ideal controller, which features perfect information on the attitude angles, there are still errors associated to the aerodynamic response of the vehicle.

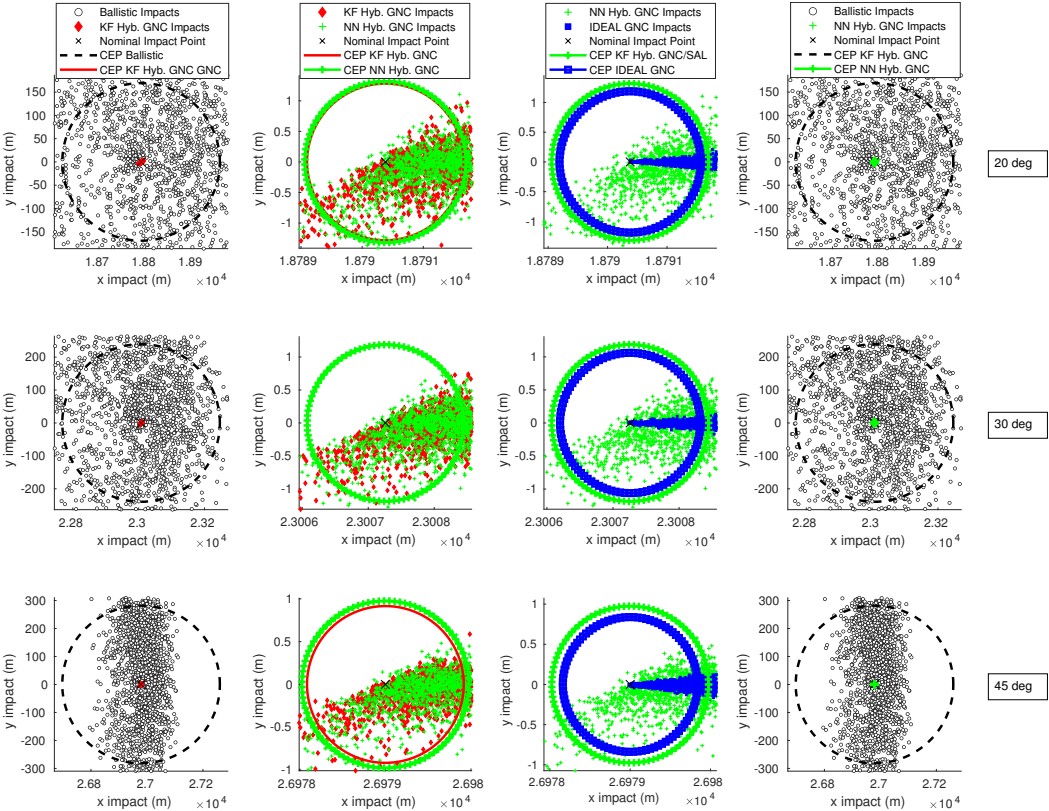

**Figure 4.** Detailed shots for different algorithms.

**Table 8.** The CEP in neural network based controlled flights for the different methods and training algorithms for 20°, 30° and 45° initial pitch angles.

|  |  | CEP | |
| :---: | :---: | :---: | :---: |
| Init. Ang. (deg) | Alg. | Method 1 | Method 2 |
| 20 | SCG | 2211.93 | 2115.80 |
| 30 | SCG | 2174.67 | 2369.82 |
| 45 | SCG | 2565.99 | 2285.93 |
| 20 | BR | 1.23 | 1.33 |
| 30 | BR | 1.19 | 1.25 |
| 45 | BR | 0.99 | 1.15 |
| 20 | LM | 1.22 | 1.31 |
| 30 | LM | 1.17 | 1.26 |
| 45 | LM | 0.95 | 0.97 |

## 5. Conclusions

A novel methodology, which depends on an innovative hybridization among several sensor signals, has been proposed. At the core of the approach, neural networks are employed to get estimations of the gravity vector, which allows avoiding the use of gyroscopes. Traditional GNSS/IMU frameworks feature little errors of up to one meter, which may imply huge mistakes in line of sight vector computation when separation to the objective is small. With the proposed approach the exactness of line of sight calculation can be improved during the terminal GNC, enhancing the accuracy at the destination point, while sensor needs are lowered.

The proposed approach employs information gathered from GNSS, acceloremeters, and a semiactive laser kit. With that information, two different neural network architectures are applied to estimate the gravity vector in order to determine the attitude of the vehicle. Three training algorithms have been addressed to tune the parameters in the neural networks. In total, six different strategies are developed for estimating the gravity vector along the trajectory. In addition, because the methodology allows for determining attitude in two ways, the information is hybridized with the aim of augmenting precision.

This innovative methodology is integrated into a two-phase guidance algorithm for aerial vehicles, which provides the required input data for the GNC system. The guidance law is founded on a constant glide angle and on a modified proportional law. The control algorithm is based on a robust and effective but simple double-input double-output device. Overall, the resulting GNC system presents excellent values regarding dispersion at the destination objective, significantly increasing the precision for noncontrolled flights, as expected, but also matching accuracy as provided by other GNC systems requiring more sensors on-board. Note that results also show good behavior of the system under uncertainty conditions. Summarizing, the developed approach, which is based on neural networks, shows that precision levels can be matched or improved as compared to other methodologies.

Future research will address the increase of the use of neural networks in other modules of GNC algorithms to further simplify the overall architecture.

**Author Contributions:** Conceptualization, R.d.C.; methodology, R.d.C. and L.C.; software, L.C.; validation, R.d.C.; formal analysis, R.d.C. and P.S.; investigation, R.d.C. and P.S.; resources, L.C.; data curation, R.d.C. and P.S.; writing–original draft preparation, L.C.; writing–review and editing, R.d.C.; visualization, R.d.C. All authors have read and agreed to the published version of the manuscript.

**Funding:** This research was funded by Project Grant F663—AAGNCS by the "Dirección General de Investigación e Innovación Tecnológica, Consejería de Ciencia, Universidades e Innovación, Comunidad de Madrid" and "Universidad Rey Juan Carlos".

**Acknowledgments:** The authors would like to thank Lieutenant Colonel Jesús Sánchez (NMT) of the National Institute for Aerospace Technology (INTA) for the solid modeling of the concept.

**Conflicts of Interest:** The authors declare no conflict of interest.

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
