# Peer review of "Applying Neural Networks in Aerial Vehicle Guidance to Simplify Navigation Systems"

_algorithms, doi:10.3390/a13120333_

Round 1

Reviewer 1 Report

In this work, the authors present an algorithm that simplifies navigation of a vehicle by approximating the map between vehicle acceleration and the gravity vector by means of neural networks. Nowadays, there are numerous applications to use Machine learning techniques to simplify complex systems or algorithms and this work lies in the trend. I like that the authors are critical of the approach they present, though there are some minor drawbacks in the manuscript:

1) The choice of an learning method, its parameter values, neural network architectures cannot be selected by only a literature review, it is always a task specific to the problem

2) I think that the authors should consider a way to present formulas to all forces, maybe in a special Appendix section

3) Formulas should be before references to them

4) Please, clearer introduce the systems of reference in 2.2. xe is used before it is defined. A picture will be helpful.

5) Am I right that the proposed algorithm gives almost the same performance quality as the Kalman filter approach? Why then you call it "improving"?

Taking into account all the above, I think that the manuscript can be accepted for publication after a minor revision.

Author Response

Dear Editor,

Herewith I am submitting a revised version of our paper Applying Neural Networks in Aerial Vehicle Guidance to Simplify Navigation Systems. We want to thank you as well as the referee for the comments and suggestions. We revised the paper according to the reviewer's comments and suggestions. In what follows, we answer the reviewer's comments in detail.

With kind regards,

Raúl de Celis

Reviewer 1:

In this work, the authors present an algorithm that simplifies navigation of a vehicle by approximating the map between vehicle acceleration and the gravity vector by means of neural networks. Nowadays, there are numerous applications to use Machine learning techniques to simplify complex systems or algorithms and this work lies in the trend. I like that the authors are critical of the approach they present, though there are some minor drawbacks in the manuscript:

The authors are grateful for these comments.

1) The choice of a learning method, its parameter values, neural network architectures cannot be selected by only a literature review, it is always a task specific to the problem

The reviewer is right. These choices depend on the problem to be solved. The authors meant that after reviewing the literature addressing learning of dynamics, two main methodologies were assumed to be applicable to this problem. With this survey, a seminal hyperparametric study was performed to determine the optimal working point, whose results suggested that there is a limit number of neurons in the intermediate layer (estimated in around 100 neurons), and over this limit, results do not get improved. This is now clearly stated in the current version of the paper. Note that this process was specific to the presented problem in the paper.

2) I think that the authors should consider a way to present formulas to all forces, maybe in a special Appendix section

We have now included them in the current version of the paper. The follow the convention in [27] (STANAG). The full expressions are showed by equations (2), (3) and (4).

3) Formulas should be before references to them

This issue has been corrected in the current version of the paper.

4) Please, clearer introduce the systems of reference in 2.2. xe is used before it is defined. A picture will be helpful.

The reviewer is right. Now, Figure 1 shows the employed reference frames.

5) Am I right that the proposed algorithm gives almost the same performance quality as the Kalman filter approach? Why then you call it "improving"?

The reviewer is right. But note the improvement is got at two different points. First, there is no need for gyros (they are needed in the Kalman filter approach), which reduces the complexity and the cost of the overall system. Second, there is no need for knowing the plant model, as it may be needed for the mechanization equations.

Taking into account all the above, I think that the manuscript can be accepted for publication after a minor revision.

The authors are grateful for these positive and constructive comments.

Reviewer 2 Report

Article is interesting, the subject is current and has useful value. In order to enhance the article quality, I suggest the following remarks be taken into account:

  • The References should be extended to include the publications that refer to the discussed subject, for instance:
    • Borkowski P., Pietrzykowski Z., Magaj J., MÄ…ka M. „Fusion of data from GPS receivers based on a multi-sensor Kalman filter” Transport Problems vol. 3, no. 4, 2008 (5-11)
    • JaroÅ›, A. Witkowska and R. Åšmierzchalski, "Data fusion of GPS sensors using Particle Kalman Filter for ship dynamic positioning system," 2017 22nd International Conference on Methods and Models in Automation and Robotics (MMAR), Miedzyzdroje, 2017, pp. 89-94, doi: 10.1109/MMAR.2017.8046804
  • The authors should add units in Table 2.
  • The choice of the number and shape of neurons as well as the amount of training and validation data selected for the presented two strategies is important for prove the suitability of the proposed approach. Please present the details of the performed hyperparametric study.
  • I suggest that for a better understanding of the paper content and for an easier implementation of the proposed algorithm it would be necessary to rewrite the section 3 by including a flowchart of the algorithm and its algorithmic presentation with all the steps that need to be taken.
  • Figure 2 seems to be insufficiently described.

Author Response

Dear Editor,

Herewith I am submitting a revised version of our paper Applying Neural Networks in Aerial Vehicle Guidance to Simplify Navigation Systems. We want to thank you as well as the referee for the comments and suggestions. We revised the paper according to the reviewer's comments and suggestions. In what follows, we answer the reviewer's comments in detail.

With kind regards,

Raúl de Celis

Reviewer 2:

Article is interesting, the subject is current and has useful value. In order to enhance the article quality, I suggest the following remarks be taken into account:

The authors are grateful for these comments.

The References should be extended to include the publications that refer to the discussed subject, for instance:

    • Borkowski P., Pietrzykowski Z., Magaj J., MÄ…ka M. „Fusion of data from GPS receivers based on a multi-sensor Kalman filter” Transport Problems vol. 3, no. 4, 2008 (5-11)
    • JaroÅ›, A. Witkowska and R. Åšmierzchalski, "Data fusion of GPS sensors using Particle Kalman Filter for ship dynamic positioning system," 2017 22nd International Conference on Methods and Models in Automation and Robotics (MMAR), Miedzyzdroje, 2017, pp. 89-94, doi: 10.1109/MMAR.2017.8046804

These references have been now added to the discussion in the current version of the paper.

The authors should add units in Table 2.

According to additional suggestions received by other reviewers, Table 2 have been suppressed.

The choice of the number and shape of neurons as well as the amount of training and validation data selected for the presented two strategies is important for prove the suitability of the proposed approach. Please present the details of the performed hyperparametric study.

Further and detailed hyperparametric study will be performed in future work to precisely determine the optimal working point but is not the objective of this research to formalize this statement. The preliminary results of this hyperparametric study suggest that there is a limit number of neurons in the intermediate layer (estimated in around 100 neurons), and over this limit, results do not get improved. This is now clearly stated in the current version of the paper.

Regarding the amount of data for training and validation for the two strategies, the paper now says that the percentage of data employed in this training is a 70% and that as it is common practice, a representative amount of data is left aside for validation purposes. In this case, a 15% of the available data corresponds to validation. In addition, the trained NN is tested with the independent data (15% of the collected data).

I suggest that for a better understanding of the paper content and for an easier implementation of the proposed algorithm it would be necessary to rewrite the section 3 by including a flowchart of the algorithm and its algorithmic presentation with all the steps that need to be taken.

A new paragraph has been added to section 3. It describes the flowchart of the process. In addition, Figure 2 has been added, which shows the general process.

Figure 2 seems to be insufficiently described.

This figure shows the ballistic trajectory that the vehicle follows for three different initial pitch angles without using the control system at all, that is, it shows the behavior of the system before implementing the improvements provided by the GNC.

Reviewer 3 Report

While the proposed idea is interesting and seemingly novel, the manuscript is fairly superficial. The authors should re-write the manuscript using more rigorous mathematical language, consistent notation, and unambiguous definitions. The authors should be more careful when defining and using the underlaying reference frames and associated coordinate systems. The meaning of the first two figures (Table 2 and Figure 1) is unclear and, in the reviewer's opinion, unnecessary. The authors should choose a standardized aerospace convention and stick to it throughout the manuscript. Moreover, the authors should better explain the motivation/rationale of their selected (design) choices. More realistic conditions should be considered for numerical investigations. It is not clear whether load factor limitations were imposed in the simulations. A plot with scattered impact points and required load factor orders should be included in the manuscript.

Author Response

Dear Editor,

Herewith I am submitting a revised version of our paper Applying Neural Networks in Aerial Vehicle Guidance to Simplify Navigation Systems. We want to thank you as well as the referee for the comments and suggestions. We revised the paper according to the reviewer's comments and suggestions. In what follows, we answer the reviewer's comments in detail.

With kind regards,

Raúl de Celis

Reviewer 3:

While the proposed idea is interesting and seemingly novel, the manuscript is fairly superficial.

The authors should re-write the manuscript using more rigorous mathematical language, consistent notation, and unambiguous definitions.

The authors have followed the convention established in reference [27] (STANAG). Now this reference has been added to the current version of the paper to clearly show internationally recognized conventions are being followed. In addition, all the involved mathematical expressions in the forces and moments have been detailed.

The authors should be more careful when defining and using the underlaying reference frames and associated coordinate systems.

The authors have reviewed the use of the definitions to avoid employing undefined concepts. In addition, Figure 1 has been added in order to clarify this aspect.

The meaning of the first two figures (Table 2 and Figure 1) is unclear and, in the reviewer's opinion, unnecessary.

The reviewer is right. The table referred to key findings in previous research and they are not vital to understand the current contributions of this paper, and the figure was generic at some point. Consequently, they have been deleted.

The authors should choose a standardized aerospace convention and stick to it throughout the manuscript.

The authors acknowledge the convention here could be weird. However, it is drawn from the Standardization Agreement at the NATO [27], which has been widely used and tested before. But at the end, the authors believe this should not be an insurmountable issue, as it is very similar to other aerospace conventions.

Moreover, the authors should better explain the motivation/rationale of their selected (design) choices.

Note the main motivation is to simplify the GNC system. This is achieved here by the following two main issues. First, there is no need for gyros (for example, they are needed in the Kalman filter approach), which reduces the complexity and the cost of the overall system. Second, there is no need for knowing the plant model, as it may be needed for the mechanization equations. This stated in the paper as follows: “The  main  contribution  of  this  scientific  research  is  the  application  of  Machine  Learning  techniques, i.e.,  Neural  Network  (NN)  algorithms,  to  hybridize  GNSS,  accelerometer  and  semi-active  laser  quadrant photo-detectors signals.  The role of the Neural Networks is to predict the gravity vector to estimate the attitude of the vehicle. Consequently, the advantage of such a hybrid system over the traditional ones, which are usually based on GNSS and IMUs, is the capability of eliminating gyros, which may be expensive and too sensitive for high demanding maneuvers and not reliable at all during the first stages of a launch.”

Regarding physical restrictions and designs of the vehicle, they are given by the Spanish air force, as this is a research based on their vehicle. Further details on the vehicle and sensors may be found in [1,2,17,28].

More realistic conditions should be considered for numerical investigations. It is not clear whether load factor limitations were imposed in the simulations.

Non-controlled shots have been validated against real data provided by the Spanish air force. then Monte Carlo simulations are performed to calculate closed-loop performance over a full range of uncertainty, as suggested by experts in the air force to model the possible uncertainty that can arise in aspects such as initial conditions, sensor information procurement, weather conditions, and thrust properties. The authors believe these are realistic settings.

Regarding load factor limitations, they were not needed in the computational experiments here. Due to the configuration of the vehicle and the tested flight, load factors never exceeded the limit. In addition, because the focus of the paper is the development and integration of a neural network in the GNC architecture, these constraints were not imposed. And, note that imposing load factor limitations would not modify the proposed approach in regards to the neural network.

A plot with scattered impact points and required load factor orders should be included in the manuscript.

A new figure has been added to the paper to show impact points.

Round 2

Reviewer 2 Report

The authors addressed my previous comments. I am satisfied with this revision.